# The Incidence and Clinical Characteristics of Interstitial Lung Disease Associated with CDK4/6 Inhibitors in Breast Cancer Patients: A Retrospective Multicenter Study

**DOI:** 10.3390/medicina61030549

**Published:** 2025-03-20

**Authors:** Nurullah İlhan, Akif Doğan, Hande Nur Erölmez, Fatih Atalah, Süleyman Baş, Servan Yasar, Hatice Odabaş, Mahmut Gümüş

**Affiliations:** 1Department of Medical Oncology, Health Science University, Sancaktepe, Şehit Prof Dr. İlhan Varank Training Research Hospital, Istanbul 34785, Turkey; drakifd@gmail.com; 2Department of Family Medicine, Health Science University, Sancaktepe, Şehit Prof Dr. İlhan Varank Training Research Hospital, Istanbul 34785, Turkey; herolmez@gmail.com; 3Department of Medical Oncology, Faculty of Medicine, Medeniyet University, Prof. Dr. Süleyman Yalçın City Hospital, Istanbul 34700, Turkey; fatihatalah@gmail.com (F.A.); mgumus@superonline.com (M.G.); 4Department of Internal Medicine, Health Science University, Sancaktepe, Şehit Prof Dr. İlhan Varank Training Research Hospital, Istanbul 34785, Turkey; suleymanbas.2012@gmail.com; 5Department of Radiology, Health Science University, Sancaktepe, Şehit Prof Dr. İlhan Varank Training Research Hospital, Istanbul 34785, Turkey; yasarservan@gmail.com; 6Department of Medical Oncology, Health Science University, Kartal Dr. Lütfi Kırdar City Hospital, Istanbul 34865, Turkey; odabashatice@yahoo.com

**Keywords:** CDK4/6 inhibitors, breast cancer, hypersensitivity pneumonitis, nonspecific interstitial pneumonia, pulmonary toxicity

## Abstract

*Background and Objectives:* CDK4/6 inhibitors (CDK4/6i) have revolutionized the treatment of hormone receptor-positive HER2 negative (HR(+)/HER2(-)) breast cancer. Despite their efficacy, interstitial lung disease (ILD) remains a rare but potentially fatal adverse effect. This study aims to evaluate the incidence and clinical characteristics of ILD associated with CDK4/6 inhibitors in breast cancer patients in Turkey. *Materials and Methods*: A retrospective multicenter analysis included 464 breast cancer patients treated with CDK4/6 inhibitors between January 2017 and April 2024. Patients receiving ribociclib or palbociclib were evaluated for the development of ILD. Radiological assessments were performed to confirm ILD and exclude other conditions. Clinical characteristics, treatment regimens, and outcomes were analyzed. *Results*: ILD was identified in 10 patients (2.1%). The average age of the affected patients was 62.5 ± 9.85 years. Hypersensitivity pneumonitis and nonspecific interstitial pneumonia (NSIP) were the most common radiological patterns. Palbociclib was implicated in six cases, while ribociclib was associated with four cases. Grade 3 pulmonary toxicity was observed in eight patients, and Grade 4 toxicity in two patients. One patient who was on palbociclib died due to ILD. No significant correlation was found between ILD and age, smoking status, lung metastases, or prior thoracic radiotherapy. *Conclusions*: The incidence of CDK4/6 inhibitor-associated ILD in Turkish breast cancer patients appears higher than previously reported in clinical trials. More robust, long-term studies are necessary to identify potential risk factors and mitigate ILD-related mortality.

## 1. Introduction

Breast cancer is the most common cancer among women worldwide, accounting for 31% of all female cancers [1]. In 2022, 2.3 million breast cancer diagnoses were made globally, with 670,000 deaths attributed to breast cancer [2]. Breast cancer remains a significant cause of morbidity and mortality worldwide. Breast cancer subtypes have been classified as luminal A, luminal B, human epidermal growth factor receptor 2 (HER2) overexpression, and basal-like (triple-negative) [3,4]. In estrogen receptor (ER)-positive breast cancer, ER pathway activation leads to upregulation of the ER–cyclin D–CDK4/6 axis, which is critical for cell cycle regulation [5]. The cyclin-dependent kinase (CDK) family consists of 20 known members, each playing a crucial role in regulating cell cycle progression, transcription, and RNA splicing [6]. These kinases function within a highly coordinated network to ensure that, during cell division, DNA is precisely replicated and equally distributed between two daughter cells. Any disruption in the regulation of cell cycle checkpoints or transcriptional control can trigger apoptosis; however, if left uncorrected, such disturbances may contribute to the development of various diseases, including cancer, neurodegenerative disorders (such as Alzheimer’s and Parkinson’s), and stroke [7,8,9]. Within the CDK family, CDK4/6 phosphorylates the retinoblastoma (Rb) protein upon activation through cyclin D binding, releasing E2F transcription factors. This event initiates the transcription of genes required for cell cycle progression, facilitating the transition from the G1 to S phase [10,11]. One of the most crucial checkpoints in the cell cycle occurs at the G1/S transition, which is predominantly controlled by the retinoblastoma protein (Rb). In its active, hypophosphorylated state, pRb binds to and inhibits E2F transcription factors, thereby preventing progression into the S phase. However, when phosphorylated by the cyclin D–CDK4/6 complex, pRb releases E2F, enabling transcription of genes necessary for DNA replication and cell cycle progression [12]. This regulatory pathway is further influenced by upstream signaling cascades, such as the PI3K/AKT/mTOR and RAS/MAPK pathways, which are activated in response to mitogenic and hormonal stimuli, including estrogen receptor (ER) activation [13]. In malignancies such as HR+/HER2- breast cancer, dysregulation of these pathways contributes to uncontrolled proliferation, and tumor progression leads to unchecked breast cancer cell proliferation. Inhibition of CDK4/6 has marked a significant milestone in breast cancer treatment. CDK4/6 inhibitors are molecularly targeted agents. By inhibiting CDKs, they prevent the formation of the CDK4/6–cyclin D complex, thereby blocking cell division and exerting antitumor activity [14]. The combination of CDK4/6 inhibitors (CDK4/6i) with endocrine therapy (ET) has been approved by the FDA as a first-line treatment regimen for patients with advanced hormone receptor-positive, HER2-negative (HR+/HER2-) breast cancer [15]. Furthermore, CDK4/6 inhibitors (CDK4/6i) are reported to be applicable as adjuvant therapy in patients with high-risk early-stage breast cancer [16]. Their positive effects on patient survival, quality of life, and favorable safety profiles have made CDK4/6 inhibitors a cornerstone of breast cancer treatment. CDK4/6 inhibitors are generally associated with mild to moderate, non-lethal adverse effects such as myelosuppression, gastrointestinal symptoms, and rash. However, they can also lead to rare but potentially fatal side effects, such as interstitial lung disease [17,18]. Interstitial lung disease (ILD) associated with CDK4/6 inhibitors is a rare (1–3.3%) but profound adverse effect. ILD is a complex medical condition with a multifactorial etiology. The mechanism of action of ILD due to CDK4/6 inhibitors is unclear. Still, it is suspected that drugs in this class may induce cellular senescence and increase tissue inflammation and fibrosis of the lung interstitium [19]. It often presents with non-specific respiratory symptoms, such as dyspnea and cough, and can be life-threatening in some instances. As a diagnosis of exclusion, there are no universally established, systematically applied criteria for its identification [20]. Assessing pulmonary toxicity related to specific therapeutic agents is particularly challenging in breast cancer patients, as many may have pre-existing lung conditions, including metastatic involvement or prior radiation-induced damage. These patients often receive multiple systemic therapies in different combinations and sequences, making isolating individual drug-related pulmonary effects difficult. Interstitial lung disease (ILD), wherein the mechanism of event onset is typically unknown, has no established preventive or therapeutic measures. Among the drugs used in the treatment of advanced breast cancer, everolimus and trastuzumab deruxtecan, both associated with a high incidence of ILD, are of particular concern. Everolimus, an mTOR inhibitor, has been reported to be linked with ILD in 18% of cases [21]. Trastuzumab deruxtecan, a HER2-targeted antibody–drug conjugate (ADC) used in the treatment of HER2-positive breast cancer, has been associated with ILD, with an incidence of 14% reported in clinical studies [22]. Although ILD associated with CDK4/6 inhibitors appears less frequent than with these drugs, it should not be overlooked.

Given the increasing use of targeted therapies, including CDK4/6 inhibitors, some of which have been associated with ILD risk, it is crucial to identify factors contributing to ILD development. This would enable proactive patient monitoring, facilitating early ILD detection and timely intervention while allowing patients to receive optimal oncologic treatment when possible.

In general, the literature indicates that anticancer drug-induced ILD is associated with advanced age, a history of prior ILD, and a smoking history [23]. However, in Turkey, the frequency of ILD development related explicitly to CDK4/6 inhibitors used in breast cancer, the impact of different CDK4/6 inhibitor molecules, and additional risk factors contributing to this frequency have not yet been identified. Our study, the first of its kind conducted in Turkey, aims to determine the incidence of CDK4/6 inhibitor-associated ILD in breast cancer patients and to characterize the clinical features of the affected patients.

## 2. Materials and Methods

### 2.1. Study Design and Data Collection

Our study is a retrospective, multicenter study in which data were collected by retrospectively reviewing archived patient files and hospital medical records. If patients had been admitted to other departments, such as intensive care units or pulmonary disease services, their medical records were meticulously examined and analyzed in collaboration with relevant specialists to ensure data accuracy and comprehensiveness. Patient data recorded included age (>55 and ≤55 years), ECOG score at diagnosis, ECOG score, adjuvant/neoadjuvant/metastatic stage chemotherapy status (before CDK4/6i initiation), presence of lung metastases related to breast cancer, other metastatic sites, smoking history, menopausal status, history of ILD, and history of thoracic radiotherapy.

### 2.2. Study Population

The study included 464 breast cancer patients aged over 18 years who received CDK4/6 inhibitors (CDK4/6i) between 1 January 2017 and 30 April 2024 for HR+/HER2 metastatic or locally advanced disease. Due to reimbursement conditions in Turkey, abemaciclib is not used for HR+/HER2- disease. Consequently, none of the patients in this study received abemaciclib. All patients included in the study were treated with ribociclib and palbociclib or received both sequentially. Those with clinical and radiological conditions that could mimic ILD (such as pneumonia, lymphangitic carcinomatosis, or radiation pneumonitis) unrelated to CDK4/6i use were excluded from the study. A comprehensive diagnostic approach was utilized to differentiate CDK4/6 inhibitor-associated interstitial lung disease (ILD) from other potential causes, mainly prior chest radiotherapy. Patient medical records were reviewed to assess prior exposure to radiotherapy, including the irradiated lung volume, dose, and time interval between treatment and symptom onset. Respiratory symptoms were analyzed in relation to CDK4/6 inhibitor initiation and other underlying pulmonary conditions, such as chronic obstructive pulmonary disease, infections, or heart failure. High-resolution computed tomography (HRCT) scans were independently evaluated by thoracic radiologists blinded to the treatment history. ILD cases were classified based on radiological patterns, where the presence of diffuse ground-glass opacities, organizing pneumonia, or hypersensitivity pneumonitis-like findings was considered more indicative of drug-induced lung injury. At the same time, fibrosis or volume loss confined to prior radiation fields suggested radiation pneumonitis. Additional diagnostic workups, including bronchoalveolar lavage (BAL) in selected cases, were conducted to exclude infectious or inflammatory causes, and serum inflammatory markers (e.g., C-reactive protein and procalcitonin) were assessed when available.

The response to CDK4/6 inhibitor discontinuation and corticosteroid therapy was also evaluated. Cases demonstrating clinical and radiological improvement following drug withdrawal, with or without corticosteroid treatment, were classified as CDK4/6i-associated ILD. In contrast, those with persistent fibrosis or progressive deterioration despite drug cessation were further assessed for radiation-induced lung injury or alternative etiologies. Finally, cases with uncertain diagnoses were reviewed in a multidisciplinary tumor board comprising oncologists, pulmonologists, and radiologists to achieve a consensus diagnosis. Conditions that could mimic ILD (such as pneumonia, lymphangitic carcinomatosis, or radiation pneumonitis) were excluded by reviewing patient files, including thoracic CT scans, sputum cultures, EBUS, bronchoalveolar lavage, and other necessary procedures. After excluding other clinical causes, an experienced thoracic oncologic radiologist radiologically evaluated ILD associated with CDK4/6 inhibitors. Radiological patterns of ILD were classified into five groups: 1. Acute interstitial pneumonia (AIP)/acute respiratory distress syndrome (ARDS). 2. Nonspecific interstitial pneumonia (NSIP). 3. Organizing pneumonia (OP). 4. Acute eosinophilic pneumonia. 5. Hypersensitivity pneumonitis (HP). The severity of ILD was graded using the universal CTCAE (Common Terminology Criteria for Adverse Events) version 5.0, established by the National Cancer Institute (NCI).

### 2.3. Ethical Approval

Ethical approval for this study was obtained from the Ethics Committee of Sancaktepe Şehit Prof. Dr. İlhan Varank Training and Research Hospital (Ethics Committee Decision No:128 Date: 17 April 2024). All human studies were conducted according to the principles outlined in the 1964 Declaration of Helsinki. As the study was designed retrospectively, written informed consent from patients was not required.

### 2.4. Statistical Analysis

IBM SPSS version 28 was used for statistical analyses. Descriptive statistical methods (mean, standard deviation, median, frequency, percentage, minimum, and maximum) were employed to evaluate the study data. The Kolmogorov–Smirnov and Shapiro–Wilk tests were used to assess the normality of quantitative data. The independent *t*-test was applied to compare two groups of normally distributed quantitative variables, while the Mann–Whitney U test was used for non-normally distributed variables. Categorical data were compared using the chi-square test and Fisher’s exact test. Kaplan–Meier (log-rank) analysis was performed for survival analysis. A *p*-value of less than 0.05 was considered statistically significant.

## 3. Results

### 3.1. Baseline Characteristics

The demographic and clinical characteristics of the patients are presented in Table 1. A total of 464 patients were diagnosed with metastatic breast cancer, with a median age of 59.06 ± 13.18 years at the initiation of CDK4/6 inhibitor therapy. Most were female (99.1%, *n* = 460), while 0.9% (*n* = 4) were male. Before CDK4/6 inhibitor therapy, 55.0% maintained an ECOG status of 0, whereas 32.3% had a status of 1, 12.1% had a status of 2, and 0.6% had a status of 3. CDK4/6 inhibitors were administered as first-line therapy in 65.1% of cases, second-line in 20.9%, and third-line or beyond in 14.0%. Among the agents used, ribociclib was prescribed to 54.5% of patients, palbociclib to 44.2%, and 1.3% received both sequentially. Endocrine therapy combined with CDK4/6 inhibitors included letrozole in 64.0%, fulvestrant in 34.7%, and other endocrine agents in 1.3%. Chemotherapy exposure before CDK4/6 inhibition treatment was observed in 51.1% of patients, with 30.2% receiving neoadjuvant chemotherapy, 16.2% adjuvant chemotherapy, and 2.6% chemotherapy in the metastatic setting (Table 1). The remaining 48.9% had no prior chemotherapy. The distribution of metastatic sites included bone (82.2%), lung (35.3%), liver (23.5%), soft tissue (11.2%), pleura (7.8%), brain (6.3%), skin (2.6%), and peritoneum (1.9%). Regarding smoking history, 71.3% were never smokers, 25.6% were former smokers, and 3.0% were current smokers. There were 10 patients with ILD associated with CDK4/6i. The median follow-up duration was 72.2 months (range: 4.1–334.8), with a mean follow-up of 92.9 ± 70.7 months (Table 1).

### 3.2. Evaluation of the Characteristics of Patients Diagnosed with Interstitial Lung Disease Associated with CDK4/6 Inhibitors

The comparison of patients based on their pneumonitis status is presented in Table 2. The incidence of ILD was higher in patients aged >55 years (80%) compared to those aged ≤55 years (20%), but this difference was not statistically significant (*p* = 0.129). Regarding CDK4/6 inhibitor selection, ribociclib and palbociclib use was similar between the ILD-positive and ILD-negative groups, with no statistically significant association (*p*-values: 0.523, 0.349, and 1.000 for ribociclib, palbociclib, and sequential use, respectively).

The rate of smoking was lower in patients with pneumonitis, which was statistically significant (*p* = 0.047). The presence of lung metastases was higher in ILD-positive patients (60%) compared to ILD-negative patients (34.8%), though this trend did not reach statistical significance (*p* = 0.099). No significant differences were detected between patients with and without pneumonitis regarding menopausal status, history of thoracic radiotherapy, chemotherapy status before CDK4/6i therapy, the line of CDK4/6i therapy, or lung metastases (*p* > 0.05). While older age, lung metastases, and prior treatment history appeared more frequently in ILD-positive patients, no single variable was an independent predictor of ILD risk. Further research is warranted to identify patient subgroups at higher risk for ILD development in the context of CDK4/6 inhibitor therapy.

Upon evaluation of the CT findings of patients with confirmed ILD, hypersensitivity pneumonitis was detected in four patients, NSIP in four patients, and AIP/ARDS in two patients (Table 3).

**Table 3 medicina-61-00549-t003:** Radiological findings and ILD subtypes in CDK4/6i-associated pneumonitis.

Patient ID	ILD Subtype	CDK4/6 Inhibitor Type	Radiological
			Findings
Patient 1	Hypersensitivity	Ribociclib	Upper lobe-
	Pneumonitis (HP)		predominant mosaic attenuation, air
			trapping,
			centrilobular
			nodules, and
			ground-glass
			opacities. **(Figure 1)**
Patient 2	Hypersensitivity	Ribociclib	Mosaic attenuation, air
	Pneumonitis (HP)		trapping,
			centrilobular
			nodules, and
			ground-glass
			opacities. Upper and middle lobe-
			predominant.
Patient 3	Hypersensitivity	Ribociclib	Bilateral lobe, air
	Pneumonitis (HP)		trapping,
			centrilobular
			nodules, and
			ground-glass
			opacities. **(Figure 2)**
Patient 4	Hypersensitivity	Ribociclib	Bilateral mosaic attenuation, air
	Pneumonitis (HP)		trapping,
			centrilobular
			nodules, and
			ground-glass
			opacities.
Patient 5	Nonspecific	Ribociclib	
	Interstitial		Bilateral
	Pneumonia (NSIP)		symmetrical
			ground-glass
			opacities, reticular densities, and
			traction
			bronchiectasis.
Patient 6	Nonspecific	Palbociclib	Bilateral
	Interstitial Pneumonia (NSIP)		symmetrical
			ground-glass
			opacities, reticular densities, and
			traction
			bronchiectasis. **(Figure 3)**
Patient 7	Nonspecific	Palbociclib	Lower lobe-
	Interstitial		predominant,
	Pneumonia (NSIP)		bilateral
			symmetrical
			ground-glass
			opacities, reticular densities, and
			traction
			bronchiectasis.
Patient 8	Nonspecific	Palbociclib	Lower lobe-
	Interstitial		predominant,
	Pneumonia (NSIP)		bilateral
			symmetrical
			ground-glass
			opacities, reticular densities, and
			traction
			bronchiectasis.
Patient 9	Acute Interstitial Pneumonia/ARDS (AIP-ARDS)	Palbociclib	Ground-glass opacities,
			consolidations, and honeycombing. **(Figure 4)**
Patient 10	Acute Interstitial Pneumonia/ARDS (AIP-ARDS)	Palbociclib	Diffuse ground-glass opacities,
			consolidations, and honeycombing.

### 3.3. Evaluation of Characteristics of Patients Diagnosed with Interstitial Lung Disease Associated with CDK4/6 Inhibitors

Among 464 patients diagnosed with breast cancer, 10 (2.1%) were found to have confirmed interstitial lung disease (ILD) associated with CDK4/6 inhibitors. The average age of patients diagnosed with ILD was 62.50 ± 9.85 years. Upon evaluation of the CT findings of patients with confirmed ILD, hypersensitivity pneumonitis was detected in four patients, NSIP in four patients, and AIP/ARDS in two patients (Table 3).

Among the 10 patients who developed ILD associated with CDK4/6 inhibitors(Table 4), the mean age was 62.5 ± 9.85 years, and 9 out of 10 patients were postmenopausal. Six patients had lung metastases, whereas all had bone metastases. Only two patients had pre-existing pulmonary comorbidities, and five had a history of prior chest radiation, both of which are recognized risk factors for ILD. The most frequently observed ILD subtype was hypersensitivity pneumonitis (HP) (4/10, 40%) and nonspecific interstitial pneumonia (NSIP) (4/10, 40%), followed by acute interstitial pneumonia/ARDS (AIP-ARDS) in two patients (20%).

Among the ribociclib cases, the drug was initiated as first-line therapy in two patients, second-line treatment in one patient, and third-line therapy in one patient. For palbociclib cases, the drug was initiated as first-line therapy in four patients, second-line therapy in one patient, and third-line treatment in one patient. In combination with ribociclib, letrozole was used in three cases and fulvestrant in one case, whereas in combination with palbociclib, letrozole was used in five cases and fulvestrant in one case (Table 5).

ILD severity was classified as grade 3 in eight patients (80%) and grade 4 in two patients (20%). Due to ILD-related complications, CDK4/6i treatment was permanently discontinued in nine patients (90%). In one patient, due to advanced age, comorbidities, and the exhaustion of other treatment options, a risk-based decision was made to reintroduce CDK4/6i therapy. Corticosteroid therapy was administered in all patients, and oxygen support was required in nine patients (90%), reflecting significant respiratory impairment. Four patients (40%) received antibiotic therapy, while one patient (10%) required intensive care unit (ICU) admission due to severe respiratory distress. Additionally, one patient (10%) developed persistent pulmonary dysfunction; mycophenolate mofetil (MMF) and infliximab were used in one case (10%) as an additional immunosuppressive strategy. The median ILD resolution time was 35.0 days (range: 21–56 days, mean: 35.2 days). ILD-related death occurred in one patient (10%), while nine patients (90%) survived.

The median time to ILD onset was 161 days (range: 73–581 days, mean: 225.1 days). (Table 5 and Figure 5). The longest duration before ILD diagnosis was 581 days, while the shortest was 73 days.

## 4. Discussion

This study is significant as it is the first in Turkey to analyze cases of ILD caused by CDK4/6 inhibitors. In this retrospective study, we identified CDK4/6 inhibitor-induced ILD in 10 out of 464 cases (2.1%). Although the rate of CDK4/6 inhibitor-induced ILD is reported to be between 1 and 1.3% in clinical studies [24], we found a higher rate in our study. This discrepancy may be attributed to several factors. Real-world data often capture a broader and more heterogeneous patient population than controlled clinical trials, where strict inclusion criteria may underrepresent patients with pre-existing pulmonary conditions, extensive metastatic disease, or prior lung radiation. Another possible explanation is that our study population included a higher proportion of elderly patients and those with lung metastases, both of which have been hypothesized as potential risk factors for CDK4/6 inhibitor-induced ILD. In clinical trials, mortality rates associated with CDK4/6 inhibitor-induced ILD are low. No fatal ILD cases were reported during the 5-year follow-up of combination therapy with palbociclib and endocrine therapy [25]. In the MONARCH-2 study for abemaciclib, two deaths (0.5%) due to pneumonitis and one death (0.1%) in the MONARCH-3 study were reported [26]. Despite early detection and intervention, in our study, one patient died (0.22% of all 464 patients receiving CDK4/6 inhibitors) due to ILD-related complications, emphasizing the potentially life-threatening nature of CDK4/6 inhibitor-associated pneumonitis. In FAERS, 96% of CDK4/6 inhibitor-induced ILD cases occurred in patients aged 50 and over; the rate of ILD caused by CDK4/6 inhibitors was approximately 2.1% in those using abemaciclib and 0.3% in those using palbociclib/ribociclib, CDK4/6 inhibitor-induced deaths were reported in 18 patients using abemaciclib, 24 patients using palbociclib, and 5 patients using ribociclib [27]. In a meta-analysis by Zhang et al., the incidence of ILD/pneumonitis associated with CDK4/6 inhibitors was 1.6% [28]. In a study conducted by Chen et al., advanced age, a history of ILD, and poor performance status were found to be associated with abemaciclib-induced mortality [29]. In the MonarchE study, a higher incidence of ILD was observed in patients who had previously received thoracic radiotherapy [30]. In our cohort, among the patients that had interstitial lung disease (ILD) associated with CDK4/6 inhibitors, 60% of patients had lung metastases, 50% had a history of chest radiation, and 20% had known pulmonary comorbidities. However, advanced age and lung metastases were observed more frequently in the group that developed pneumonitis; although numerically higher values were present, they did not reach statistical significance. This may be attributed to the rarity of ILD and the limited number of CDK4/6 inhibitor-induced ILD cases in our cohort. Interestingly, in our study, smoking history was found to be statistically significantly lower in patients who developed pneumonitis. This finding appears paradoxical, as smoking is generally associated with an increased risk of lung injury and ILD due to pre-existing lung damage (smoking-related interstitial lung diseases (SR-ILD)) [31]. One possible explanation is a demographic factor specific to our study population. Most ILD-related studies have been conducted in developed countries, where smoking prevalence among women is generally higher. In contrast, in Turkey, smoking rates among women are relatively lower, which may have influenced the statistical distribution of smoking history in our cohort. As a result, the overall number of smokers in our study may have been insufficient to reveal a clear association between smoking and pneumonitis risk, given the small number of ILD cases. Additionally, selection bias may have played a role, as patients with a history of smoking might have undergone stricter pulmonary evaluations before treatment initiation, potentially leading to early detection or modification of therapy in high-risk individuals. In our study, CDK4/6 inhibitor therapy was permanently discontinued in 9 patients (90%) due to ILD-related complications. This high discontinuation rate underscores the severity of ILD as an adverse event associated with CDK4/6 inhibitors, aligning with previous reports highlighting pulmonary toxicity as a significant limitation of treatment continuation [32]. Given the potentially life-threatening nature of ILD, early identification and prompt intervention are crucial to mitigate its impact. Despite the risks, in our study of one patient, rechallenge with CDK4/6 inhibitors was considered due to advanced age, comorbidities, and the exhaustion of alternative treatment options. The decision was based on the patient’s complete resolution of pneumonitis symptoms within 30 days, supporting the hypothesis that timely corticosteroid intervention and drug discontinuation may allow for symptom resolution in select cases [33]. Consequently, therapy was cautiously resumed with a dose reduction, corticosteroid support, and close monitoring to mitigate the risk of recurrence. Although mild pulmonary symptoms persisted, the patient tolerated the reintroduction of CDK4/6i without further severe complications. This case suggests that, under stringent clinical surveillance, a rechallenge strategy may be feasible for a carefully selected patient population. However, considering the potential for ILD recurrence, a thorough risk–benefit assessment should guide individualized treatment decisions. According to the existing literature, in some cases where CDK4/6 inhibitors cause grade 3/4 adverse effects, the treatment is often discontinued, and once the adverse effect resolves, either the same drug is reintroduced at a reduced dose or a switch to a different CDK4/6 inhibitor is made. This approach is particularly evident in cases of hepatotoxicity, where switching from one CDK4/6 inhibitor to another has been reported without the recurrence of the same adverse effects [34,35]. However, there are currently insufficient data and evidence to apply a similar strategy to cases of CDK4/6 inhibitor-induced interstitial lung disease (ILD). Further studies are warranted to establish standardized guidelines for the safe rechallenge of CDK4/6 inhibitors in patients with a history of ILD. In our study, the median time to ILD onset was 161 days. The longest duration before ILD diagnosis was 581 days, while the shortest was 73 days. This considerable variability in ILD onset aligns with previous reports, which have demonstrated that ILD associated with CDK4/6 inhibitors can manifest at any stage of treatment, from as early as a few weeks to more than a year after therapy initiation [28,36]. It should also be noted that the diagnosis of CDK4/6 inhibitor-related ILD may be delayed due to its often insidious onset and the attribution of symptoms to comorbidities or disease metastasis. The slow progression of respiratory symptoms, such as dyspnea and cough, can lead to misinterpretation as underlying pulmonary conditions, infection, or cancer progression rather than drug-induced toxicity [37]. This diagnostic challenge underscores the importance of heightened clinical awareness and routine monitoring for ILD in patients receiving CDK4/6 inhibitors to ensure early detection and timely intervention. However, our findings suggest that ILD can develop even later, reinforcing the need for continuous monitoring throughout therapy. Additionally, the literature indicates that ILD associated with anticancer agents exhibits a highly variable onset, necessitating ongoing vigilance beyond the early phases of treatment [37]. Given this variability, clinicians must maintain a high level of suspicion for ILD at all stages of CDK4/6 inhibitor therapy. Continuous monitoring, early recognition of respiratory symptoms, and prompt intervention remain critical to mitigating ILD severity and improving patient outcomes. Identifying potential predictive factors for early- versus late-onset ILD could help refine monitoring strategies and personalize risk assessment. Further research is warranted to elucidate the underlying mechanisms influencing the timing of ILD onset and to develop standardized monitoring protocols for patients receiving CDK4/6 inhibitors. These results should be interpreted cautiously, and further research is needed to explore the potential mechanisms underlying this association. In light of these findings, risk factors for CDK4/6 inhibitor-induced ILD remain unclear, emphasizing the need for more extensive, long-term studies to better identify potential predictors.

### Limitations

This study has several limitations. First, the number of patients who developed ILD in our cohort was limited to only 10, making it difficult to draw firm conclusions regarding specific risk factors for ILD in breast cancer patients. More significantly, multi-center studies with a higher number of ILD cases are needed to better define these associations. Second, although this study was conducted as a multi-center study across hospitals in Turkey, the findings may still not be fully generalizable to breast cancer patients worldwide due to potential regional differences in genetic predisposition, environmental factors, and healthcare practices. Future studies with a more diverse and international patient population are required to validate our results in different healthcare settings. The most important feature of the study is that it is the first of its kind in Turkey. Ribociclib and palbociclib, CDK4/6 inhibitors available in Turkey, have become the first-line option in HR(+)/HER2(-) advanced breast cancer patients without visceral crisis when combined with aromatase inhibitors or endocrine therapy (ET), and their use has rapidly increased. The incidence of ILD associated with CDK4/6 inhibitors in Turkey is unknown, making this study important as the first conducted in the country.

## 5. Conclusions

Our study indicates that the incidence of ILD associated with CDK4/6 inhibitors may be high. This study, the first in Turkey, provides real-world insights into CDK4/6 inhibitor-induced ILD. We identified an ILD incidence of 2.1%, higher than that reported in clinical trials, possibly due to differences in patient populations, including a higher proportion of elderly patients and those with lung metastases. The onset of ILD varied widely, occurring between 73 and 581 days after treatment initiation, reinforcing the need for continuous monitoring throughout therapy. Although CDK4/6 inhibitor-induced ILD is generally rare, its impact on treatment outcomes is significant. In our study, 90% of affected patients permanently discontinued therapy due to ILD-related complications, and one patient (0.22%) died despite early intervention. While lung metastases and advanced age were more common in ILD cases, these differences did not reach statistical significance, highlighting the challenge of identifying apparent risk factors. Interestingly, we observed a lower smoking history among patients who developed pneumonitis, a finding that appears paradoxical and may reflect demographic characteristics or selection bias. Further research is needed to clarify this association.

Additionally, we demonstrated that, in select cases, rechallenge with a reduced dose and corticosteroid support may be a feasible approach under strict clinical monitoring. Given our cohort’s limited number of ILD cases, our findings should be interpreted with caution. More significantly, multi-center studies with diverse patient populations are needed to better define risk factors, optimize early detection strategies, and develop standardized management guidelines for CDK4/6 inhibitor-induced ILD.

## Figures and Tables

**Figure 1 medicina-61-00549-f001:**
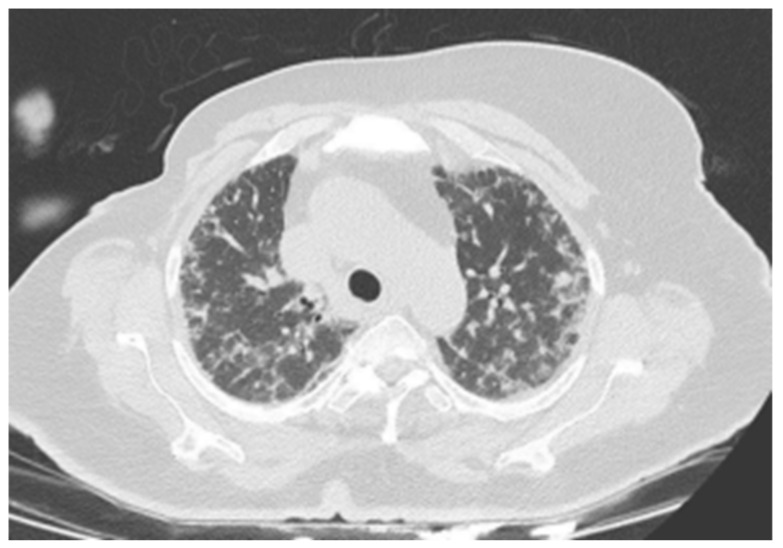
There are widespread micronodular and patchy ground-glass areas in the upper zones of both lungs, and they were evaluated in favor of hypersensitivity pneumonitis.

**Figure 2 medicina-61-00549-f002:**
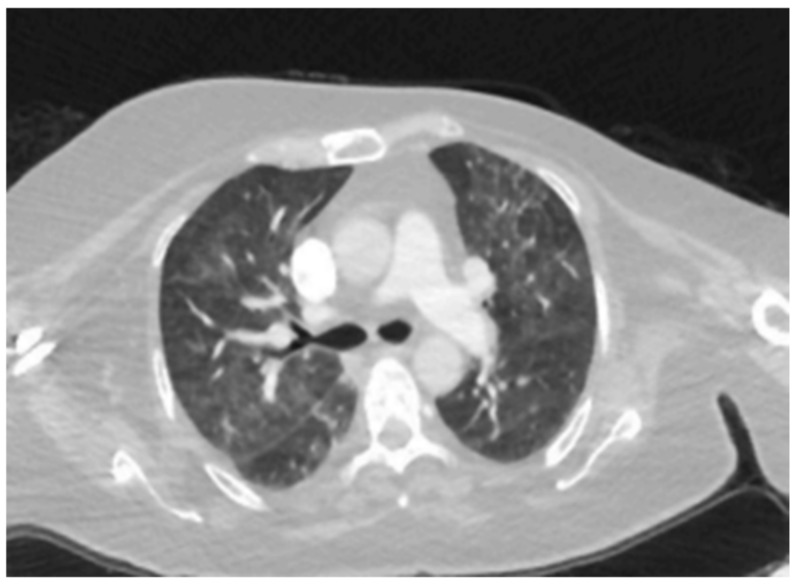
Scattered patchy ground-glass densities in both lungs.

**Figure 3 medicina-61-00549-f003:**
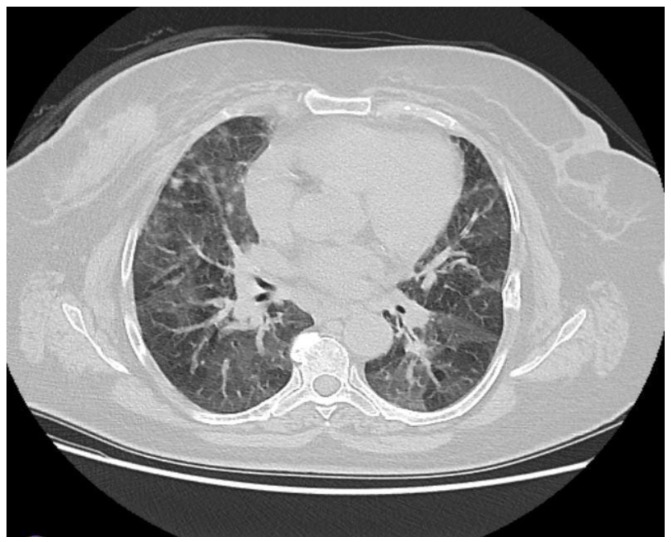
Symmetrical ground-glass areas in both lungs, predominantly in the basal regions, were evaluated in favor of nonspecific interstitial pneumonia.

**Figure 4 medicina-61-00549-f004:**
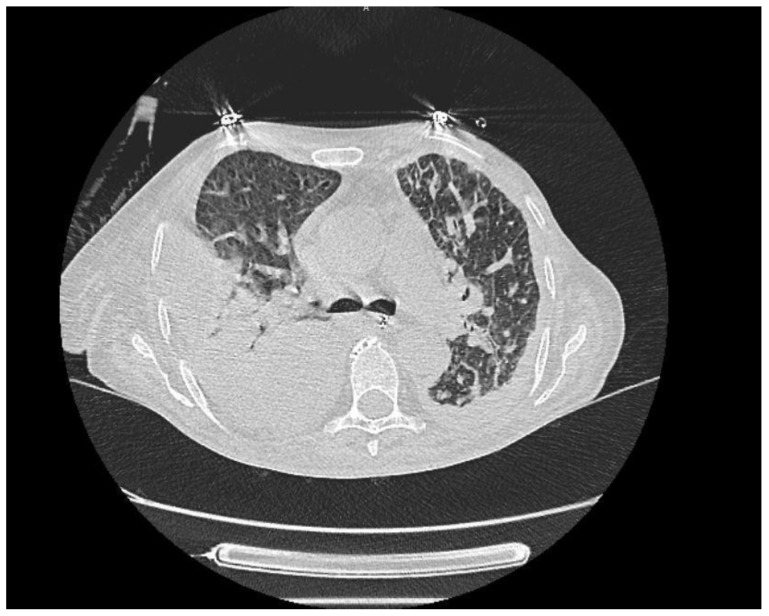
More prominent consolidated and ground glass areas and interlobular septal thickenings in the scattered posterior sections of both lungs. These are also accompanied by bilateral pleural effusion. This case was evaluated in favor of ARDS, along with the clinical findings.

**Figure 5 medicina-61-00549-f005:**
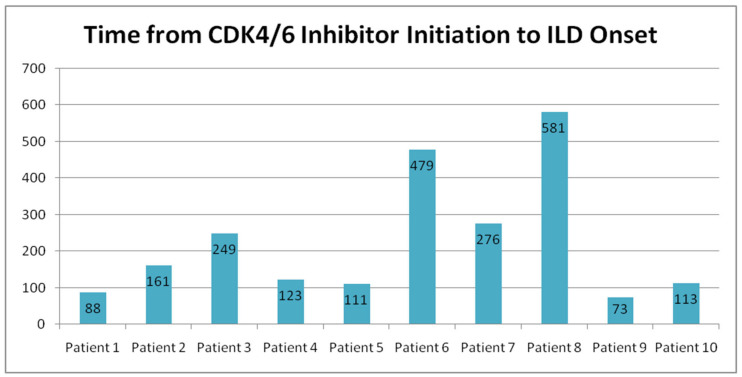
Time from CDK4/6 inhibitor initiation to the onset of ILD.

**Table 1 medicina-61-00549-t001:** Demographic and clinical characteristics of patients.

	Parameter	Value
**Metastatic Breast Cancer Patients**		(*n* = 464)
**Gender**	Male Patients	4 (0.9%)
	Female Patients	460 (99.1%)
**Age (Mean ± Standard Deviation)**	Age at Initiation of CDK4/6i Therapy (Years)	59.06 ± 13.18
**ECOG Performance Status**		
	**At Diagnosis**	
	0	309 (66.6%)
	1	132 (28.4%)
	2	23 (5%)
	**Before CDK4/6i Therapy**	
	0	255 (55%)
	1	150 (32.3%)
	2	56 (12.1%)
	3	3 (0.6%)
**Line of CDK4/6i Therapy**		
	1	302 (65.1%)
	2	97 (20.9%)
	3 and later	65 (14.0%)
**Type of CDK4/6 Inhibitor**		
	Ribociclib	253 (54.5%)
	Palbociclib	205 (44.2%)
	Sequential Use of Ribociclib and Palbociclib	6 (1.3%)
**Endocrine Therapy Used with CDK4/6 Inhibitors**		
	Letrozole	297 (64%)
	Fulvestrant	161 (34.7%)
	Other	6 (1.3%)
**Chemotherapy Status**		
**Before CDK4/6i Therapy**		
	Neoadjuvant Chemotherapy	140 (30.2%)
	Adjuvant Chemotherapy	75 (16.2%)
	Metastatic Chemotherapy	12 (2.6%)
	No Chemotherapy	227 (48.9%)
**Metastatic Sites Before CDK4/6i Therapy**		
	Bone	384 (82.2%)
	Lymph Node	206 (44.4%)
	Lung	164 (35.3%)
	Liver	109 (23.5%)
	Soft Tissue	52 (11.2%)
	Pleura	36 (7.8%)
	Brain	29 (6.3%)
	Skin	12 (2.6%)
	Peritoneum	9 (1.9%)
**Smoking Status**		
	Current Smoker	14 (3%)
	Former Smoker	119 (25.6%)
	Never Smoked	331 (71.3%)
**ILD Associated with CDK4/6i**		
	Yes	10 (2.2%)
	No	454 (97.8%)
**Follow-up Duration (Months)**	Median (Min–Max)	72.2 (4.1–334.8)
	Mean ± Standard Deviation	92.9 ± 70.7

**Abbreviations:** CDK4/6i: CDK4/6 inhibitors, ILD: interstitial lung disease.

**Table 2 medicina-61-00549-t002:** Evaluation of risk factors for interstitial lung disease associated with CDK4/6 inhibitors.

	Interstitial Lung Disease Associated with CDK4/6 Inhibitor		
Parameter	No (*n* = 454)	Yes (*n* = 10)	*p*-Value
**Gender (Female)**	450 (99.1%)	10 (100%)	1.000 a
**Age at Initiation of CDK4/6i Treatment (Years) (Mean ± Standard Deviation)**	56.9 ± 12.6	53 ± 13.3	0.251 b
**Age at Start of CDK4/6i Treatment**			
≤55 Years	200 (44.1%)	2 (20%)	0.129 c
>55 Years	254 (55.9%)	8 (80%)	
**CDK4/6i Treatment**			
Ribociclib	249 (54.8%)	4 (40%)	0.523 c
Palbociclib	199 (43.8%)	6 (60%)	0.349 c
Sequential Use of Ribociclib and Palbociclib	6 (1.3%)	0 (0%)	1.000 a
**Pre-existing ILD**			
Yes	4 (0.9%)	0 (0%)	0.152 a
No	450 (99.1%)	10 (100%)	
**Smoking Status**			
Current Smoker	14 (3.1%)	0 (0%)	0.047 * c
Former Smoker	119 (26.2%)	0 (0%)	
Never Smoked	321 (70.7%)	10 (100%)	
**Menopause Status Before CDK4/6i**			
Postmenopausal	321 (70.7%)	9 (90%)	0.183 a
Premenopausal	133 (29.3%)	1 (10%)	
**History of Radiation to the Chest Area**			
Yes	228 (50.2%)	5 (50%)	0.989 c
No	226 (49.8%)	5 (50%)	
**Chemotherapy Status Before CDK4/6i Therapy**			
Yes	232 (51.1%)	5 (50%)	0.945 c
No	222 (48.9%)	5 (50%)	
**Line of CDK4/6i Treatment**			
1st Line	296 (65.2%)	6 (60%)	0.858 c
2nd Line	95 (20.9%)	2 (20%)	
3rd and Subsequent Lines	63 (13.9%)	2 (20%)	
**Lung Metastasis**			
Yes	158 (34.8%)	6 (60%)	0.099 c
No	296 (65.2%)	4 (40%)	

**Abbreviations:** CDK4/6i: CDK4/6 inhibitors, ILD: interstitial lung disease. **Footnotes:** * Statistically, the significance level was lower than 0.05. “a” Fisher’s exact test, “b”: independent *t* test, “c”: chi-square test.

**Table 4 medicina-61-00549-t004:** Clinical characteristics, treatment, and outcomes of ILD cases.

Pt ID	Age (Yrs)	CDK4/6i	Tx Line	PS	Lung Metastasis	Bone Metastasis	Smoking Status	Menopause	Pulm Comorbidity	Chest RT	Chemo Prior CDK4/6i	**CerbB2**	**ILD Type**	**ER/PR Status**
1	59	R	1	1	Y	Y	N	Post	N	N	N	0	HP	ER 100%/PR +
2	62	R	2	2	N	Y	N	Post	N	Y	Y	0	HP	ER 100%/PR (+)
3	67	P	1	1	Y	Y	N	Post	N	Y	Y	0	HP	ER 80%/PR (+)
4	58	R	1	0	Y	Y	N	Post	Y	N	N	0	HP	ER 90%/PR (+)
5	68	P	2	0	N	Y	N	Post	N	Y	Y	0	NSIP	ER 100%/PR (+)
6	49	P	1	0	Y	Y	N	Post	N	Y	Y	0	NSIP	ER 100%/PR (+)
7	70	P	1	1	N	Y	N	Post	N	N	N	1	NSIP	ER 95%/PR (+)
8	63	P	3	2	Y	Y	N	Post	Y	N	N	1	NSIP	ER 100%/PR (+)
9	81	P	1	2	N	Y	N	Post	N	N	N	1	AIP-ARDS	ER 98%/PR (+)
10	48	R	3	0	Y	Y	N	Pre	N	Y	Y	1	AIP-ARDS	ER 50%/PR (-)

Abbreviations: Pt ID: Patient ID, Age (yrs): age in years, CDK4/6i: CDK4/6 inhibitor (R: ribociclib, P: palbociclib), ET: endocrine therapy (L: letrozole, F: fulvestrant), Tx Line: treatment line; PS: performance score, Lung Met: lung metastasis, Bone Met: bone metastasis, Smoke: smoking status, Meno: menopause status (Post: postmenopausal, Pre: premenopausal); Pulm Comorb: pulmonary comorbidity, Chest RT: history of chest radiation, Chemo Prior CDK4/6i: prior chemotherapy before CDK4/6i, CerbB2: CerbB2 score; ILD Type: interstitial lung disease type (HP: hypersensitivity pneumonitis, NSIP: nonspecific interstitial pneumonia, AIP-ARDS: acute interstitial pneumonia–acute respiratory distress syndrome); Tox G: pulmonary toxicity grade (G3: grade 3, G4: grade 4), ER/PR Status: estrogen receptor/progesterone receptor status. Table 5 summarizes the treatment approaches, ILD severity, therapeutic modifications, and clinical outcomes of 10 patients diagnosed with CDK4/6 inhibitor (CDK4/6i)-related interstitial lung disease (ILD).

**Table 5 medicina-61-00549-t005:** ILD case details—management and clinical outcomes of 10 patients.

PtID	CDK4/6i Tx	ET	Time to Onset (Days)	ILDG.	Management	Steroid	Supportive Tx	**Resolution (Days)**	**Death**
1	Ribociclib (1st line)	Letrozole	88	3	Discontinued	Yes	Oxygen support	21	No

2	Ribociclib (2nd line)	Fulvestrant	161	3	Discontinued	Yes	Oxygen support,	24	No
antibiotics
3	Palbociclib (1st line)	Letrozole	249	3	Discontinued	Yes	ICU admission, oxygen support,	N/A	Yes
antibiotics, MMF, infliximab
4	Ribociclib (1st line)	Letrozole	123	3	Discontinued	Yes	Oxygen support	35	No
5	Palbociclib (2nd line)	Fulvestrant	111	3	Discontinued	Yes	Oxygen support	37	No
6	Palbociclib (1st line)	Letrozole	479	4	Discontinued	Yes	Oxygen support	42	No
7	Palbociclib (1st line)	Letrozole	276	3	Discontinued	Yes	Oxygen support, MMF, persistent pulmonary dysfunction	N/A	No
8	Palbociclib (3rd line)	Letrozole	581	4	Discontinued	Yes	Oxygen support	28	No
9	Palbociclib (1st line)	Letrozole	73	3	Discontinued	Yes	Oxygen support, antibiotics	45	No
10	Palbociclib (3rd line)	Letrozole	113	3	Resumed with Dose Reduction	Yes	Oxygen support	30	No

## Data Availability

The data presented in this study are available on request from the corresponding author.

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
