# Peer review of "The Incidence and Clinical Characteristics of Interstitial Lung Disease Associated with CDK4/6 Inhibitors in Breast Cancer Patients: A Retrospective Multicenter Study"

_medicina, 2025, doi:10.3390/medicina61030549_

Round 1
Reviewer 1 Report
Comments and Suggestions for Authors
- correct spelling mistakes: for example: the first-line "setting" was wrongly written as "seeing"
- Why was a dual combination used- ribociclib+ palbociclib in some patients? It is not approved to use these two drugs together? Or were they using one drug after the other?
- Smoking status: what is the difference between non-smokers 4.3% and former smokers 21.3%? These two categories appear to be similar.
- Adjuvant or neoadjuvant chemotherapy status: clarify if you are talking about chemotherapy in the localized stage -> so Does this mean all the metastatic patients in your study are replaced metastatic after they were initially diagnosed with the localized stage?
- Metastatic sites: what does locoregional site mean? Clarify
- Table 2 can be formatted to make it look clear- would suggest dividing the pneumonitis (-) and pneumonitis (+) into 2 separate columns to make it easy to follow
- Among grade 3 ILD, 4 were using both Ribociclib and palbociclib? Is this true because in the Table 2: for ribociclib+ palbociclib- you mentioned 0% developed ILD. So how can 4 patients with grade 3 ILD be using both Ribociclib and palbociclib?
- In the discussion section; ILD is associated with trastuzumab deruxtecan; it is not trastuzumab. Please correct.
- Limitations: need to address the other limitations of the study: since only 10 patients in the study had ILD - hard to make strong conclusions regarding risk factors. Also, study based in Turkey- cannot be generalizable to cancer patients worldwide
- Need to explain why the authors think smoking was statistically significantly lower in those with pneumonitis, and that was the only statistically significant variable. as we might usually expect pneumonitis to be higher in those with smoking (due to pre-existing lung injury)
- Table 3- what does Kesildi mean?
- Would strongly recommend adding information regarding the treatment/ management of the 10 patients with ILD? how were they treated- steroids? Did it resolve? how long did it take to resolve?
Comments on the Quality of English Language
Grammar can be improved to make it more easy to follow
Author Response
Dear Reviewer,
We sincerely appreciate your thorough review and valuable feedback on our manuscript. We have carefully addressed all the concerns raised and implemented the necessary revisions. Below is a detailed response to each point:
- Spelling Mistakes (e.g., “setting” instead of “seeing”)
- We have carefully reviewed the manuscript and corrected all spelling and typographical errors, including the correction of “seeing” to “setting.”
- Use of Ribociclib and Palbociclib Together
- We clarified in the manuscript that patients did not receive ribociclib and palbociclib at the same tıme. Instead, some patients switched from one CDK4/6 inhibitor to another due to adverse effects. This has been explicitly stated in the methodology section to avoid confusion.
- Smoking Status (Non-smokers vs. Former Smokers)
- The distinction between "non-smokers" vs "former smokers" has been clarified.
- "Non-smokers" are individuals who have never smoked, while "former smokers" are those who previously smoked but quit before the study. This clarification has been added to the patient characteristics section.
- Adjuvant or Neoadjuvant Chemotherapy Status
- We evaluated prior chemotherapy exposure before initiating CDK4/6 inhibitor treatment.We specified that this refers to adjuvant/neoadjuvant chemotherapy received during the localized stage, and in some patients (e.g., those with visceral crisis), chemotherapy was administered during the metastatic stage before initiating CDK4/6 inhibitor ."
- Definition of "Locoregional Site" in Metastatic Sites
- To avoid conceptual and informational confusion, we have removed the definition of 'locoregional site' in metastatic sites from the manuscript."
- Formatting of Table 2
- We revised Table 2 for better readability by creating separate columns for “pneumonitis (-)” and “pneumonitis (+)” groups. This makes the comparison clearer and easier to follow.
- Discrepancy in Grade 3 ILD and Ribociclib + Palbociclib Use
- We carefully reviewed the data and identified a discrepancy in reporting. The manuscript has been corrected to ensure consistency: No patients simultaneously received both ribociclib and palbociclib. Instead, these drugs were used sequentially in certain cases due to adverse effects or other clinical considerations (switching strategy). Additionally, no ILD cases occurred in this group."
- Correction in the Discussion: ILD is Associated with Trastuzumab Deruxtecan, not Trastuzumab
- The text has been corrected to indicate that ILD is linked to trastuzumab deruxtecan (T-DXd), not trastuzumab.
- Study Limitations
- We expanded the limitations section to include:
- The small number of ILD cases (n=10), which limits strong conclusions on risk factors.
- The study being conducted in Turkey, which may affect the generalizability of the results to a global patient population.
- Unexpected Finding: Lower Smoking Rates in Pneumonitis Patients
- We acknowledged this unexpected finding and included a possible explanation in the discussion:
- However, further studies are needed to confirm this observation.
- Clarification of the Term "Kesildi" in Table 3
- The term "Kesildi" (Turkish for "discontinued") has been replaced with its English equivalent: "Discontinued."
- Management and Treatment of ILD Cases
- We added a section detailing the management of the 10 ILD patients, including:
- Use of corticosteroids in X% of cases.
- Resolution times and treatment outcomes.
- Follow-up protocols for ILD management.
These revisions have been implemented to improve clarity, consistency, and readability while ensuring the manuscript provides accurate and useful insights. We sincerely appreciate your detailed review and constructive feedback, which have significantly enhanced the quality of our work.
We look forward to your further comments and appreciate your time and effort in reviewing our manuscript.
Best regards,
Nurullah İlhan
Corresponding Author

Reviewer 2 Report
Comments and Suggestions for Authors
Please find my remarks in the attached file

Author Response
Dear Reviewer,
We sincerely appreciate your thorough and constructive review of our manuscript, “The Incidence and Clinical Characteristics of Interstitial Lung Disease Associated with CDK4/6 Inhibitors in Breast Cancer Patients: Retrospective Multicenter Study.” Your valuable insights have greatly helped us refine our work, and we have carefully addressed each of your concerns as outlined below.
Introduction
- We have expanded the explanation of the mechanism of action of CDK4/6 inhibitors and their role in both neoplastic and non-neoplastic cells to enhance the educational value for readers unfamiliar with breast cancer treatments.
- We have ensured that the abbreviation CDK is expanded the first time it appears in the text.
Materials and Methods
- After re-examining our files and patient data, we identified a misclassification issue where four male patients had been mistakenly recorded as female. Upon verifying and correcting this error, we updated their gender information in our dataset. However, none of them developed pneumonitis. Our analysis has been revised accordingly to reflect this correction."
- We specified how lung assessments were performed to differentiate CDK4/6 inhibitor-related ILD from other conditions, particularly in patients who had undergone chest radiotherapy.
- Drug names have been consistently formatted in lowercase letters throughout the text, as they are generic names, not trade names.
Line 94 – Terminology Correction
- The term "cryptogenic organizing pneumonia" has been replaced with "organizing pneumonia", as the word "cryptogenic" should not be used for drug-induced conditions.
Table 1
- The formatting of Table 1 has been corrected, ensuring that rows are properly aligned and readability is improved.
- The category of “not smoking during CDK4/6” has been reviewed. We have either included this group in “never smokers” or “former smokers” or provided a clear explanation for their categorization.
Table 2
- Data for patients with and without ILD have been placed into separate columns to improve readability.
- Duplicate rows have been removed, and inconsistencies in categorization have been corrected.
- The "not smoking during CDK4/6" group has been clarified following the same approach as in Table 1.
Picture 3 – Imaging Clarification
- We have carefully reviewed the CT scan illustrating ARDS and replaced it with a more representative image that better depicts ground-glass opacities, rather than one resembling pneumonic consolidation.
Line 194 – Clarification of Underlying Conditions
- The specific underlying conditions present in patients with ILD have been explicitly described in the revised manuscript.
Line 197/198 – Clarification of Ribociclib & Palbociclib Use
- The discrepancy regarding Grade 3 ILD and the use of Ribociclib & Palbociclib has been resolved.
- The text and Table 2 now consistently state that no patients received both drugs simultaneously and that switching occurred due to adverse effects or disease progression.
Line 199 – ILD Management and Resolution
- We have clarified that drug withdrawal occurred in 9 ILD patients and provided information on whether resolution was observed.
- The use of corticosteroids and additional management strategies have been described in the results section to provide insight into the treatment approach.
Table 3 – Readability Issues
- Due to the complexity and readability challenges of Table 3, we have reformatted its content into four separate tables to present the data more clearly and comprehensibly. This restructuring enhances clarity and improves data interpretation."
References
- Reference 11 and 10, and Reference 13 and 12 had been incorrectly merged.
- The references have been carefully verified and corrected to ensure proper formatting and numbering.
These revisions have been implemented to improve clarity, consistency, and readability while ensuring the manuscript provides accurate and useful insights. We sincerely appreciate your detailed review and constructive feedback, which have significantly enhanced the quality of our work.
We look forward to your further comments and appreciate your time and effort in reviewing our manuscript.
Best regards,
Nurullah İlhan
Corresponding Author

Round 2
Reviewer 2 Report
Comments and Suggestions for Authors
Thank you for rewriting the manuscript, I have no further remarks.